# Insula Connectivity Abnormalities Predict Impulsivity in Chronic Heroin Use Disorder: A Cross-Sectional Resting-State fMRI Study

**DOI:** 10.3390/brainsci13111508

**Published:** 2023-10-25

**Authors:** Yan Zhang, Xiao Zhong, Yongcong Shao, Jingjing Gong

**Affiliations:** 1Department of Aviation Psychology, Air Force Medical Center, People’s Liberation Army (PLA), Beijing 100142, China; demondesmile@126.com; 2School of Psychology, Beijing Sport University, Beijing 100084, China; zhongx2022@bsu.edu.cn; 3Department of Medical Psychology, Second Medical Center, PLA General Hospital, Beijing 100853, China

**Keywords:** heroin use disorder, trait impulsivity, insula

## Abstract

Patients with heroin use disorder (HUD) often exhibit trait impulsivity, which may be an important factor in and a good predictor of addiction. However, the factor structure of HUD trait impulsivity (motor, attentional, and nonplanning) and its neural correlates are not yet known. A total of 24 male volunteers with HUD and 16 healthy control volunteers were recruited for this cross-sectional study. The Barratt Impulsiveness Scale (BIS-11) and resting-state functional magnetic resonance imaging (rs-fMRI) were employed using the insula as a seed point in an effort to understand the association between trait impulsivity and its intrinsic factors and functional connectivity (FC) between the insula and the whole brain. The HUD group in this study exhibited higher total trait impulsivity scores, motor impulsivity, and nonplanning impulsivity than the control group. Changes in FC between the right insula and the lateral occipital cortex and the right angular gyrus were significantly positively correlated with total trait impulsivity scores, motor impulsivity, and nonplanning impulsivity, whereas changes in the FC between the left insula and the left superior frontal gyrus and left frontopolar brain region were significantly negatively correlated with trait impulsivity. Thus, the insula may serve as an important biomarker for identifying trait impulsivity and its intrinsic factor structure in patients with HUDs.

## 1. Introduction

Heroin use disorder (HUD), a relatively common substance use disorder (SUD), is a chronic relapse disorder characterized by compulsive substance use [1,2]. Notably, SUDs and addictive behaviors in general are closely related to trait impulses [3,4,5]. Research suggests that trait impulsivity increases individual vulnerability to addictive behaviors [6], is a strong predictor of both addiction and persistent drug addiction status [7,8], and is strongly associated with elevated relapse rates and the risk of treatment failure [9,10].

Trait impulsivity is a personality trait that precedes events relating to the regulation of the self and is usually reflected through self-reports [11]. Individuals exhibiting trait impulsivity tend to respond quickly and in an unplanned manner to internal or external stimuli without regard for negative consequences [12]. Previous studies have shown patients with SUD to have higher trait impulsivity [10]. For example, SUD and gambling addicts exhibit similar high impulsivity or inhibitory impairment in terms of trait impulsivity, response inhibition, and impulsive decision making simultaneously [13]. And because trait impulsivity in patients with SUDs is directly associated with family inheritance [14], studies have found that high impulsivity or inhibitory deficits in childhood and adolescence have been shown to positively predict the onset and severity of substance abuse problems in adulthood [15,16]. Moreover, patients with SUDs possess similar high-impulsivity traits to their healthy non-drug-using siblings, and their response inhibition was significantly worse than that of control volunteers without SUDs [17,18].

However, trait impulsivity is a multidimensional and complex structure, and the relationship between its intrinsic components and factor structure and addictive behavior remains unclear. It is crucial to clarify which aspects of factor structure are important contributors to susceptibility to SUDs and help in accurate prediction and grading [19,20]. Some studies have suggested that all these components may have a stable relationship with substance abuse [21]. However, the fact that the overall structure of trait impulses has been reported more in current studies and the factor structure has been reported relatively less means that a unified consensus is currently lacking. Barratt proposed a three-factor model of trait impulsivity based on medical, psychological, behavioral, and social models [22]. This model classifies impulsive traits into three factors: motor impulse (motor), attentional impulse (or cognitive impulse), and lack of planning impulse (no planning) [22,23]. Motor impulsivity is manifested as an unthinking action, attentional impulsivity is manifested as attentional and cognitive instability that produces rapid decision making, and nonplanning impulsivity is manifested as a lack of self-control and future planning [22,23]. Although some research suggests that all factors may be stably related to substance abuse, the specific relationship of each factor to addictive behaviors may differ [23,24]. For example, Chen et al. [23] found that motor impulsivity scores predicted changes in treatment motivation during treatment in patients with SUD, whereas attentional impulsivity and nonplanning impulsivity scores predicted changes in craving scores during treatment. Furthermore, Nery et al. [24] found that bipolar disorder patients with SUD plus drug use disorders presented statistically higher nonplanning impulsivity than bipolar disorder patients with SUD alone. Therefore, based on this model, this study first explored the relationship between the intrinsic components and factor structure of trait impulses and addictive behaviors.

The trait impulsivity present in SUDs before substance use has been found to correlate with structural abnormalities in the prefrontal-striatal pathway [25], and abnormal manifestations of this pathway have been associated with inhibitory control of behavior. Moreover, their siblings who had never used drugs such as cocaine showed similar changes in brain region activation and gray matter volume [17,18]. Since trait impulsivity is a susceptibility marker for substance addiction [6], characterized by its genetically linked endophenotype to familial, that is, the ventral striatum-medial prefrontal cortex (VS-mPFC) circuits that are biologically based on functional (inhibitory control) and structural abnormalities, this leads to aberrant supra- and infra-control decreases that increase impulsivity and lead to substance abuse risk. However, the association of these features with brain structure is also shown in the general population that does not use addictive substances [26]. This implies that the abnormal neuroanatomical structures and activation abnormalities that have been observed in addicts may not be the only factors contributing to addiction.

Since motivation drives individuals to explore internal and external environments and select adaptive behavior and habit changes [27], some researchers have focused on the role of motivational factors. They found that dysfunction of the motivation–habit pathway may be an important neurophysiological mechanism contributing to addiction [6,28]. This pathway contains neural loops for processing and seeking driven by values such as reward or punishment, as well as automated cognitive and behavioral adaptive changes [29]. The insula is a key brain region that supports endoreception and is also involved in addiction-related processes [30]. It has been found that the insula can modulate substance use motivation through direct projections to the ventral striatum and orbitofrontal cortex (OFC) [31,32]. Results from Deng and colleagues pinpoint specific functions of the anterior insular cortex → the brainstem nucleus tractus solitarii circuit for selectively controlling motivational vigor and suggest that motivation is in part regulated by top-down regulation of dopamine signaling by the insula anterior cortex [33]. This validates the researcher’s inference. The insula has also been implicated in impulsivity [34]. For instance, in a rat gambling task, injecting methamphetamine into the insula caused rats to choose a high-risk, high-reward option [35]. These results suggest that the insula may play an important role in trait impulsivity and addictive behavior. However, the relationship between abnormal insular function and the intrinsic components and factor structures of trait impulsivity remains unclear.

Since functional magnetic resonance (fMRI) has a strong spatial sensitivity and is more likely to reflect changes in brain activity over time in the location of brain regions, this study was designed to examine trait impulsivity and their factor structures (motor impulsivity, attentional impulsivity, and nonplanning impulsivity) in patients with HUD using the Barratt Impulsiveness Scale (BIS-11) questionnaire in combination with resting-state fMRI (rs-fMRI). The insula was also used as a seed point to investigate functional connectivity (FC) abnormalities and their relationships with trait impulsivity in patients with HUD. Based on previous studies [10,36], it was hypothesized that the heroin group would have higher scores on trait impulsivity and each of its factors than the control group, and that abnormalities in insula FC would be significantly and positively correlated with trait impulsivity scores.

## 2. Materials and Methods

### 2.1. Participants

In this cross-sectional study, 24 male heroin-dependent volunteers were recruited (age = 33.29 ± 5.842 years, all right-handed) from compulsory drug treatment centers (Beijing, China) with reference to Li et al. [37]. All heroin-dependent participants were included in the experimental group after being reviewed by 2 psychiatrists based on their basic information, as well as physical examination data, and identified through a structured interview. The average duration of drug use among participants was 6.67 ± 3.712 years, and the average heroin dose is 0.904 g/day. A total of 16 healthy male adult volunteers matching in terms of age and educational level to the heroin-dependent volunteer group were recruited as a control group (age = 29.13 ± 7.898 years, all right-handed).

The specific criteria for heroin volunteers included (1) meeting the DSM-5 diagnostic criteria for heroin addiction, with at least three months of abstinence from antipsychotics with the exception of occasional use of smoking; (2) having a history of heroin use for at least two years; (3) experience of withdrawal from heroin for at least four weeks; (4) a minimum of sixth-grade education and normal grade intelligence; (5) no history of other drug consumption (alcohol, marijuana, and so on) except for cigarettes; and (6) no use of any psychiatric medication for at least three months.

The specific eligibility criteria for healthy volunteers were as follows: (1) no history of heroin or other drug use; (2) no history of head injuries or mental illness; and (3) no cardiovascular diseases.

### 2.2. Research Tools

Trait Impulsivity. The BIS-11, developed by Patton et al. in 1995, is the best tool for measuring impulsivity, as per consensus [22]. The BIS has since been revised and updated to version 11 (BIS-11), which contains 30 items across three main dimensions (attentional impulsivity, motor impulsivity, and nonplanning impulsivity) [38]. The scale rates participants’ feelings, reactions, and agreeableness on a five-point scale. The criteria are “1” for never or rarely; “2” for occasionally; “3” for from time to time; and “4” for always or almost always. In this study, the Cronbach’s α coefficient for this scale was 0.729.

### 2.3. Experimental Procedure

Before the experiment, participants completed the Barratt Impulsiveness Scale and the Personal Information Questionnaire. Then, after ensuring that the participants were in good health and met the safety standards of the MRI experiment, they were escorted by a trained scanner into the MRI suite to complete the resting state scan with their eyes closed. The body, especially the head, was kept still throughout the scan. The duration of the scan was approximately 10 min.

### 2.4. Magnetic Resonance Imaging (MRI) Data Acquisition

All neuroimaging data were collected using a 3-T Signa General Electric (GE) scanner (Signa HD, Milwaukee, WI, USA) with a standard GE quadrature head coil. Conventional anatomical images were performed using a small-angle excitation fast gradient echo sequence (T1WI-FLAIR) with transverse axial scans covering the whole brain. And the scanning parameters were as follows: time to echo (TE) = 23 ms; repetition time (TR) = 2300 ms; thickness = 5 mm; slice skip = 1 mm; slice = 24; matrix = 256 × 256; and field-of-view (FOV) = 24 × 24 cm. When using high-resolution three-dimensional T1-weighted fast spoiled gradient recalled (3D-T1-FSPGR) echo image sequencing to capture the structure of the whole brain for the subsequent superimposition of functional images, the parameters used were as follows: TE = 4.8 ms; TR = 10.4 ms; flip angle = 15; thickness = 1 mm; slice = 140; matrix = 256 × 256; and FOV = 24 × 24 cm. Functional images were acquired using a single-shot echo planar imaging gradient echo sequence with the following parameters: TE = 25 ms; TR = 2000 ms; flip angle = 90; thickness = 5 mm; slice skip = 1 mm; slice = 20; matrix = 64 × 64; and FOV = 24 × 24 cm.

### 2.5. RS-fMRI Preprocessing

The MATLAB R2021b platform (MathWorks, Inc., Natick, MA, USA), Statistical Parametric Mapping software (SPM12, https://www.fil.ion.ucl.ac.uk/spm/software/spm12/ (accessed on 20 July 2023)), and CONN toolbox [39] were used to preprocess the resting-state functional images. The preprocessing process included the following steps (Figure 1).

(1) The first ten time points of the resting-state datasets were discarded because of the initial MRI signal instability and to allow for adaptation to the scanning procedure [40]. (2) To eliminate differences in the acquisition time of each layer image, slice timing was applied to each image. (3) Head motion correction. In order to control the effect of head movements on the signal due to fatigue or respiration and blood pulsation, which are inevitable for participants during the scanning process, the head movement correction algorithm performs model alignment for rotation around the coordinate axis and translation in the direction of the coordinate axis, so that the anatomical position of the brain corresponding to a given pixel point is the same at different moments of time, even if the images at each point in time are aligned [40]. (4) Spatial normalization was applied to avoid difficulties in subsequent group-level analyses, owing to differences in the size of the participants’ heads. The average functional image space of individual subjects was normalized according to the Montreal Neurological Institute (MNI) standard space and the normalization parameters obtained during the segmentation process to resample the functional images with a voxel size of 3 mm × 3 mm × 3 mm. (5) Smoothing can reduce random noise in MRI and increase the signal-to-noise ratio. In addition, by averaging many voxels, the smoothing process increases the likelihood of normally distributed data and improves statistical power [41]. Therefore, we used a 6 mm full-width half-height Gaussian kernel to smooth the image. (6) Detrending eliminates the tendency to systematically increase or decrease the signal over time, owing to instrument instability. (7) Regression covariables. We adopted the Fisher 24 head motion parameters and added the first derivative and its square to the original six head motion parameters to minimize head motion noise [42]. We also implemented the regression of cerebrospinal fluid and white matter signals to reduce the impact of noise, such as that generated by breathing and heartbeats [43]. (8) Finally, the frequency band range of 0.01 to 0.1 Hz was used for filtering to remove the effects of high-frequency signals such as respiration and heartbeat as well as some low-frequency drift [44].

Before further processing, the quality of the preprocessed images was checked. Five participants with head movements >2 mm or 2° were excluded. The spatial normalization of the functional images was checked individually, and the alignment of the remaining 36 participants was consistent with expectations. Finally, all functional and structural images of the 36 participants were examined for abnormalities, with none found. Thus, 36 participants (23 heroin-dependent volunteers) were included in the following data analysis.

### 2.6. Seed-Based Analysis of Whole-Brain FC

FC measures the time-domain correlation between spatially separated brain regions, and the time-series correlation coefficients of brain regions with functional connectivity were significantly higher than those of non-functionally connected brain regions, even though they were spatially separated from each other [45]. The seed-based analysis is essentially a model-based approach where we can select a seed point or ROI and find the linear correlation of that seed-point region with all other voxels throughout the brain to obtain a seed-based FC map. The simplicity, interpretability, and directness of this technique makes it an excellent method to use for studying rs-fMRI FC [45]. Whole-brain resting state FC (rsFC) analysis was carried out using seed-based FC through the Conn toolbox. Seed-based analysis is a model-based approach that requires selecting a region of interest (ROI) as a seed point and then correlating the ROI with the whole brain voxels to obtain an FC map based on the seed point. The insula (left insula, MNI coordinates 55, 133, 75; right insula, MNI coordinates 129, 132, 74) was used as the ROIs, which were derived from “atlas” templates in CONN, and after extracting and removing the covariates (cephalomotor parameters, cerebrospinal fluid, white matter, and age) using multivariate linear regression [46] and performing filtering from 0.01 to 0.1 Hz to calculate the linear correlation between this seed point region and the whole-brain voxels, the correlation coefficient r values of ROI and other voxel time series of the whole brain were obtained. The r values were then transformed into z values close to a normal distribution using Fisher’s Z-transform to obtain the final transformed FC images. The two independent samples t-test was then used to determine the brain regions with differential FC between groups, controlling for age. Results presented by BrainNet Viewer (www.nitrc.org/projects/bnv/ (accessed on 20 July 2023)). The significance criteria for multiple corrections were set as multiple corrections at the uncorrected voxel level, *p* < 0.01, and corrected cluster level, *p*-FDR < 0.05.

### 2.7. Behavioral Data Analysis

#### 2.7.1. *T* Test

First, the data were checked for conformity to a normal distribution using the Shapiro–Wilk test and quantile-quantile (Q-Q) plots, and the behavioral data were largely normally distributed (see Appendix A). An independent samples t-test was then used to examine whether the scores of the two groups were different, and a raincloud plot was drawn. The above two steps were accomplished by the “stats”, “car”, and “ggplot2” packages in R (www.r-project.org (accessed on 20 July 2023)). In addition, in order to make the results more reliable, we used the JASP software (https://jasp-stats.org/ (accessed on 20 July 2023)) to supplement the existing null hypothesis significance testing (NHST) with the Bayesian factor (BF_10_) [47]. The BF_10_ indicates the ratio of the degree of support for the alternative hypothesis (i.e., there is a difference) to the degree of support for the null hypothesis (there is no difference) in the data. A value between 0 and 1 indicates that there is no evidence to support the establishment of any hypothesis; a value between 1 and 3 indicates that there is weak evidence to support the establishment of the alternative hypothesis; and a value greater than 3 indicates that there is more than moderately strong evidence to support the alternative hypothesis, with the greater the value, the stronger the evidence [48].

#### 2.7.2. ROI-Based Correlation Analysis of FC and Behavioral Data

To investigate the relationship between resting state FC (rsFC) and trait impulsivity, we extracted the change in FC value (averaged Fisher’s Z-transformed value) by subtracting the whole-brain FC matrix of the control group from the HUD group, which was formed based on the seed-based analysis of whole-brain FC. We then used Pearson correlation analysis to calculate the relationship between the change values of FC and the behavioral data. To further understand the relationship between the change value of FC and the behavioral data, we used multiple regression analysis to calculate the predictive effect of the change value of FC on the behavioral data by using the values in the correlation analysis that had a linear correlation as predictor variables. Pearson’s correlation analysis was performed using the “corrplot” and “ggpubr” packages in R, and linear regression analysis was performed using SPSS 25.0 (www.ibm.com/cn-zh/spss (accessed on 20 July 2023)).

## 3. Results

### 3.1. Behavioral Results

Independent samples *t*-tests revealed higher total impulsivity scores in the HUD group than in the control group (*t* = 2.744, *p* = 0.009, Cohen’s d = 0.890, BF_10_ = 5.286) and higher scores on the dimensions of motor impulsivity and nonplanning impulsivity (*t* = 2.130, *p* = 0.040, Cohen’s d = 0.691, BF_10_ = 1.788; *t* = 2.260, *p* = 0.030, Cohen’s d = 0.733, BF_10_ = 2.207). However, the difference between the two groups’ scores on the attentional impulsivity dimension was not significant (*t* = 1.626, *p* = 0.112, BF_10_ = 0.877; Figure 2). This suggests that both trait impulsivity and associated action and nonplanning impulsivity were higher in the HUD group than in the normal population.

### 3.2. Resting-State FC Results

The whole-brain FC characteristics of the HUD and control groups were examined under controlled age conditions, using the bilateral insula as the seed point. The results are presented in Table 1 and Figure 3 and Figure 4. We found that the FC between the insula and the three clusters in the HUD group was significantly different from that in the control group. Compared to the control group, the HUD group exhibited enhanced FC between the right insula and the lateral occipital cortex and right angular gyrus (voxel threshold: *p <* 0.01, cluster threshold: *p*-FDR < 0.05). In addition, the FC between the left insula and left superior frontal gyrus, as well as the left frontal pole and right superior frontal gyrus were diminished in the HUD group compared to the control group (voxel threshold: *p* < 0.01, cluster threshold: *p*-FDR < 0.05). The results showed an asymmetry between the left and right lateral insula.

### 3.3. Association between Insula FC and Trait Impulsivity

Correlation analyses are shown in Figure 5 and Figure 6, where changes in the FC between the right insula and the lateral occipital cortex and the right angular gyrus were significantly and positively correlated with the total BIS-11 score, motor impulsivity, and nonplanning impulsivity (*r* = 0.42, *p* = 0.010; *r* = 0.47, *p* = 0.003; *r* = 0.33, *p* = 0.045, respectively) and were not significantly correlated with attentional impulsivity. Furthermore, changes in FC between the left insula and left superior frontal gyrus and left frontopolar brain region were significantly negatively correlated with total BIS-11 scores (*r* = −0.34, *p* = 0.037) and were not significantly correlated with attentional impulsivity, motor impulsivity, or nonplanning impulsivity, whereas changes in FC between the left insula and right superior frontal gyrus were not significantly correlated with total BIS-11 scores, attentional impulsivity, motor impulsivity, or nonplanning impulsivity. This suggests that higher impulsivity traits in heroin-dependent individuals are associated with enhanced FC between the right insula and the lateral occipital cortex and the right angular gyrus and reduced FC between the left insula and left superior frontal gyrus and left frontopolar brain region.

Regression equations were constructed using trait impulsivity in individuals with HUD as the outcome variable and right and left insular FC (significantly correlated with trait impulsivity, FC between the right insula and the lateral occipital cortex and the right angular gyrus and FC between the left insula and left superior frontal gyrus and left frontopolar brain region) as the predictor variables (Table 2). The results showed that the Durbin–Watson coefficient was 1.839, indicating that there was no significant autocorrelation among the participants in the sample. FC between the right insula and the lateral occipital cortex and the right angular gyrus significantly positively predicted trait impulsivity (*β* = 0.436, *t* = 2.162, *p* = 0.043), whereas FC between the left insula and left superior frontal gyrus and left frontopolar brain region was not a significant predictor of trait impulsivity (*β* = −0.269, *t* = −1.476, *p* = 0.198). And it can explain 21.7% of the variance in variance.

## 4. Discussion

The purpose of this study was to understand the relationship between impulsivity characteristics and brain FC in patients with long-term HUD. Significantly higher total trait impulsivity, motor impulsivity, and nonplanning impulsivity were found in the HUD group than in the control group. In addition, when exploring the abnormal features of brain FC in patients with HUD using the insula as a seed point, asymmetries were found in the FC of the left and right insula; specifically, higher trait impulsivity in patients with HUD was associated with enhanced FC in the right insula and weakened FC in the left insula. Moreover, changes in right insula FC were also significantly and positively correlated with two-factor structures of trait impulsivity (motor and nonplanning).

### 4.1. Characterization of Trait Impulsivity in HUD

Previous studies have found that patients with long-term HUD often exhibit impulsivity, a trait that may be an important factor in and predictor of addiction [4,7]. The behavioral results of this study are consistent with the idea that total trait impulsivity is higher among HUD individuals than non-HUD individuals. Moreover, further interesting results were found when exploring the relationship intrinsic between impulsivity factors (motor, attentional, and nonplanning impulsivity) and HUD. The results obtained showed that the HUD group scored higher on both the motor and nonplanning impulsivity dimensions; however, the difference in scores between the two groups on the attentional impulsivity dimension was not statistically significant. Several studies have found differences in the relationships between motor impulsivity, nonplanning impulsivity, attentional impulsivity, and other behaviors or traits [49,50]. For example, among the Big Five personality traits, attentional impulsivity has been found to be positively correlated with neuroticism but not with extraversion, whereas motor impulsivity was positively correlated with extraversion but not with neuroticism [51]. Attentional impulsivity may be related to depression [52], whereas motor and nonplanning impulsivity may be related to antisocial personality and childhood trauma [53]. This study provides evidence that the relationship between the three factors of motor, nonplanning, and attentional impulsivity and addictive behaviors is also somewhat variable and further clarifies that within the factorial structure of trait impulsivity, motor, and nonplanning impulsivity are important contributors to susceptibility to HUD.

### 4.2. Relationship between Abnormal Insula Functional Connectivity and Impulsivity in HUD

The biomarkers of trait impulsivity and structural abnormalities of their factors in HUD individuals were further explored from the perspective of functional brain connectivity. First, we found enhanced FC between the right insula and the lateral occipital cortex and right angular gyrus in the HUD group compared to the control group. The lateral occipital cortex is primarily thought to be the area responsible for visual object processing and visual information integration and is associated with visual attention, a fundamental function of the external attention system [54,55]. The angular gyrus is located in the posterior part of the inferior parietal lobe, where multisensory (auditory, visual, and somatosensory) information converges and is associated with multiple cognitive domains such as memory [56,57] and visuospatial attention [58,59]. From a functional brain perspective, this result demonstrates that patients with HUD may have higher sensation seeking [12,59].

Moreover, the results of the correlation analysis between changes in the FC of the right insula with the lateral occipital cortex and right angular gyrus and trait impulsivity further revealed that higher sensation seeking and impulse traits in patients with HUD were related to the FC of the right insula. Researchers have suggested that the insula as a whole is more inclined to induce subjective feeling states in both cognition and motivation than the amygdala, which is responsible for impulse and automatic emotion processing [36,60]. Thus, the insula or its functional connections may be more relevant to emotional temperament, which identifies a stable biological or genetic “core” in an individual’s emotional response tendencies toward emotional and cognitive activities in an integrated personality [36]. More detailed evidence was also obtained by disentangling the factor structure of trait impulsivity and refining the link between intrinsic trait impulsivity factors and right insular FC. Changes in FC of the right insula were found to be significantly and positively associated with motor and nonplanning impulsivity. This is similar to the findings of Dalley and Ersche [4] that the insula is associated with the delayed discounting of sexual impulses. This result suggests a link between the FC of the left insula and a lack of impulsive behavior and self-control.

In addition, it was also found that unlike the abnormal FC of the right insula, the left insula in the HUD group had reduced FC with the left superior frontal gyrus, right superior frontal gyrus, and left frontal pole. Abnormalities in the strength of connections in the right superior frontal gyrus and right insula have previously been found in patients with schizophrenia, and these abnormalities may be related to information processing deficits [61]. Additionally, both the superior frontal gyrus and frontal pole are associated with cognitive control and executive function [62,63]. Patients with SUD show activation of the supramarginal gyrus during response inhibition-related tasks. The supramarginal gyrus is also associated with impaired response inhibition in addicts, and its abnormal activation leads to higher action impulses in addicted individuals [64]. This suggests that patients with HUD experience impaired superior frontal gyrus function after long-term drug use, leading to impaired inhibitory function and increased impulsive behavior. However, the results of the correlation analysis in this study showed that changes in the FC of the left insula, the superior frontal gyrus, and left frontopolar brain regions were significantly negatively correlated with total impulsivity scores. Conversely, changes in the FC of the left insula and right superior frontal gyrus were not linearly correlated with total impulsivity scores, nor were they linearly related to attentional impulsivity, motor impulsivity, and nonplanning impulsivity. Numerous studies have found that 5-hydroxytryptamine (5-HT) in the prefrontal cortex plays an important role in regulating an individual’s impulsivity [65,66] and that elevated concentrations of 5-HT reduce the level of compulsive cocaine-seeking in mice, which in turn prevents the onset of addiction [67]. This further supports our findings, suggesting that trait impulsivity may precede addiction [18]. Further exploration of the dynamic link between the left insula and suprafrontal gyrus and trait impulses is needed.

The results of this study notably reflect functional laterality in the cerebral hemispheres [68], that is, abnormal FC status of the different lateral insula in patients with HUD. This result provides further evidence for the existence of significant lateralized differences in the insula of the human brain [69]. Previous studies have focused on left–right differences in the insula in the areas of internal sensation [70], taste [71], and fear [72]. This study adds to the existing research evidence on insula-biased laterality differences in addiction and personality trait domains. Moreover, the present results further support the contention that dysfunction of the motivation–habituation pathway may be an important neurophysiological mechanism leading to addiction [6,28]. In the future, it may be possible to treat addictive behaviors by targeting the insula through neuromodulation techniques. It also helps us to develop targeted addiction prevention programs or public health campaigns to reduce risk factors for addiction related to impulsivity.

### 4.3. Limitations and Research Perspectives

There are some limitations in this study. First, it has been shown that gender factors play an important role in impulsive and risky behaviors [73], but in this study, due to the sampling condition, our participants were all male, which means the generalizability of the results will be affected. Future studies should consider expanding the participant pool to include a more diverse demographic, such as females and individuals with varying cultural backgrounds, to enhance the generalizability of the findings. Secondly, factors such as socio-economic status, emotion, and parameter settings of pre-processing also affect research reliability and validity, and further optimization of the experimental process is also needed in the future to exclude the interference of irrelevant factors as much as possible. Third, the ROI we chose was the entire insula region, but the functions of different subregions varied [34], so further refinement of brain regions is needed in the future to explore the role of functional connectivity in different regions of the insula in predicting trait impulsivity. We also need to further explore, at the molecular level, the biomarkers and neurobiological basis of trait impulsivity, thus providing a more comprehensive neurostructural perspective [67]. Fourth, the sample size of this study was small, and future studies could be conducted through multi-site collaborations to increase the diversity and size of the sample. Finally, although this study explored the status of trait impulsivity and its intrinsic factors in patients with HUD and its association with insular FC using behavioral and rs-fMRI data, these results are still correlational, meaning it is not yet possible based on this evidence to determine whether trait impulsivity triggers substance use or whether long-term substance use shapes trait impulsivity. Future intervention or longitudinal studies are needed to investigate the causal relationship and neurobiological mechanisms (such as dopamine pathways) between these two factors. For example, a follow-up assessment was conducted to track changes in HUD trait impulsivity over time and its dynamic relationship with the insula (or other brain regions FC) and addictive behaviors.

## 5. Conclusions

Although the fact that patients with HUDs exhibit trait impulsivity has been recognized as an essential factor in and predictor of addiction, existing research has paid little attention to the factor structure of trait impulsivity associated with HUDs. This study found higher total trait impulsivity, as well as motor and unplanned impulsivity in patients with HUD using behavioral measures and rs-fMRI. Moreover, higher trait impulsivity in patients with HUD was associated with increased FC of the right insula and decreased FC of the left insula. These findings suggest that the insula may serve as essential biomarkers for identifying trait impulsivity in patients with HUD. By focusing on the structure of trait impulsivity factors, it helps to accurately predict and personnel subgroup levels.

## Figures and Tables

**Figure 1 brainsci-13-01508-f001:**
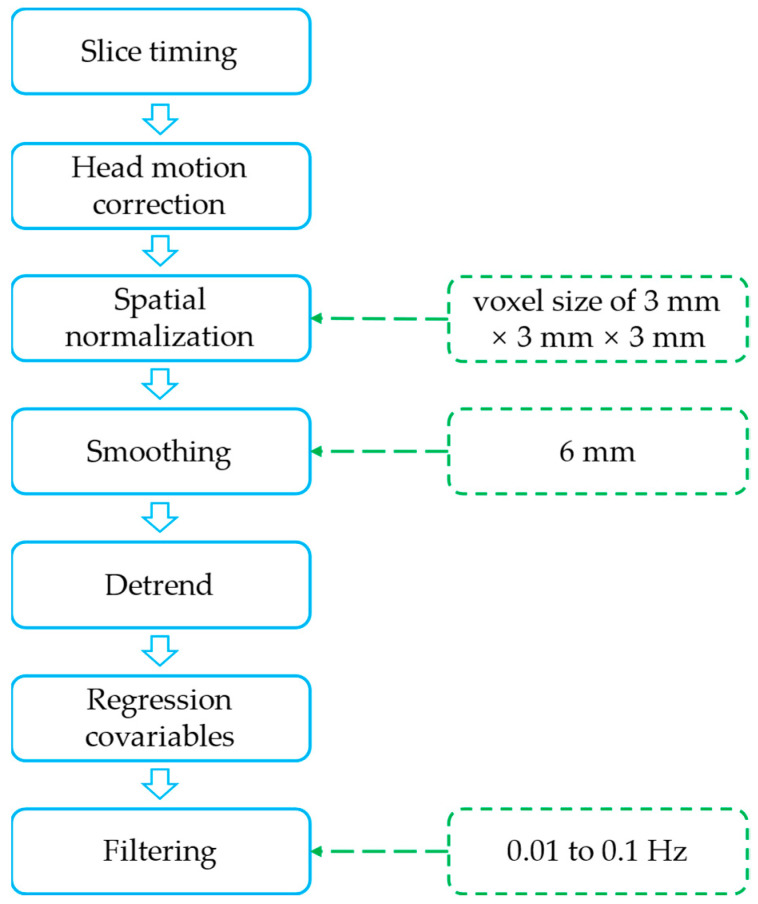
The preprocessing process.

**Figure 2 brainsci-13-01508-f002:**
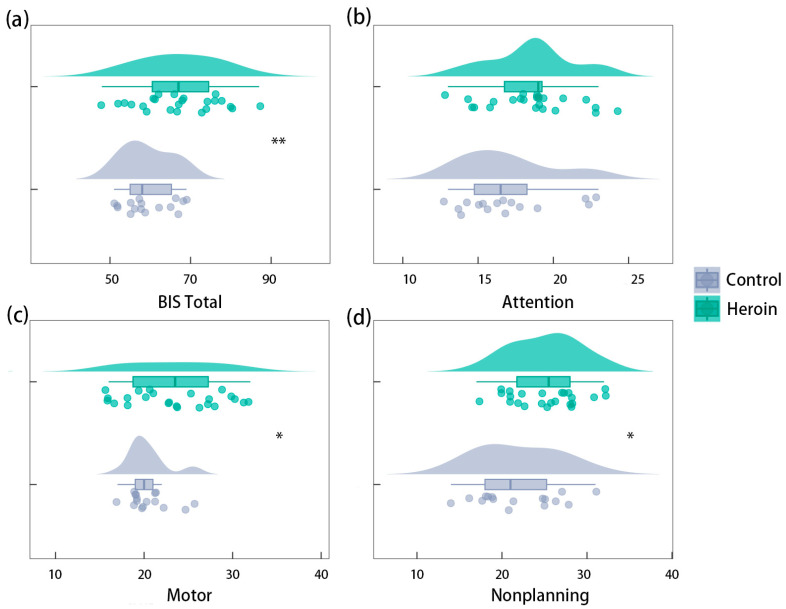
Between-group differences in BIS-11 scores. (**a**) Total BIS-11 scores were higher in the HUD group than in the control group; (**b**) motor dimension scores were also higher in the HUD group than in the control group; and (**c**) nonplanning behavior dimension scores were higher in the HUD group than in the control group; however, (**d**) the difference between the two groups was not significant in the dimension of attention. * means *p* < 0.05, ** means *p* < 0.01.

**Figure 3 brainsci-13-01508-f003:**
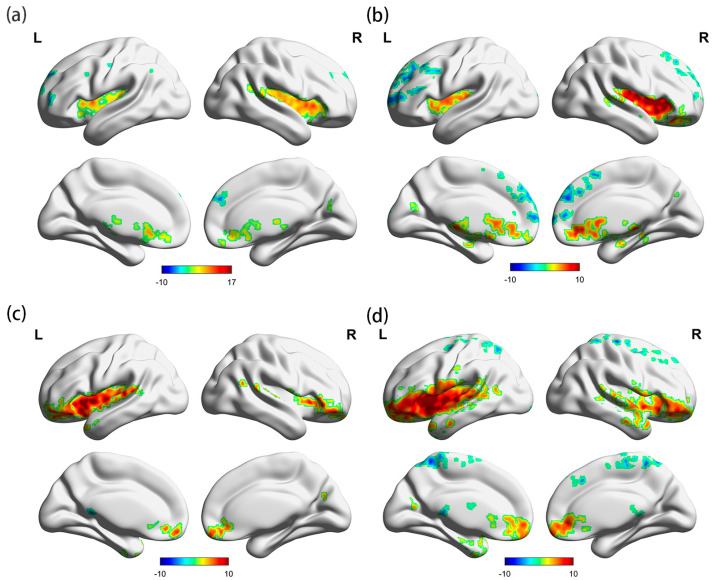
Results of the one-sample *t*-test analysis with the insula as the seed point. (**a**) One-sample *t*-test analysis of the control group when using the right insula as the seed point; (**b**) one-sample *t*-test analysis of the HUD group when using the right insula as the seed point; (**c**) one-sample *t*-test analysis of the control group when using the left insula as the seed point; and (**d**) one-sample *t*-test analysis of the HUD group when using the left insula as the seed point. Voxel threshold: *p* < 0.01, cluster threshold: *p*-FDR < 0.05. L, left, R, right.

**Figure 4 brainsci-13-01508-f004:**
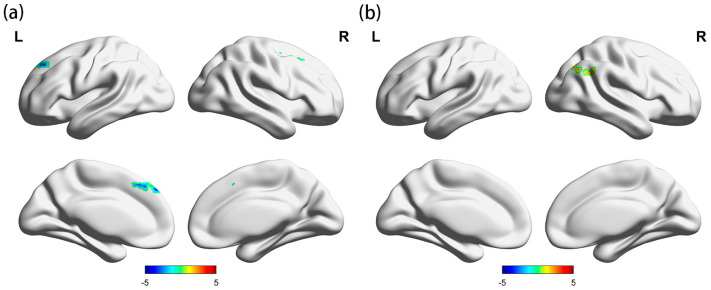
Differences in bilateral insula FC between the two groups. (**a**) FC between the right insula and the lateral occipital cortex and the right angular gyrus was enhanced in the HUD group compared with the control group, and (**b**) FC between the left insula and the left superior frontal gyrus and between the left frontal pole and the right superior frontal gyrus was diminished in the HUD group compared with the control group. Voxel threshold: *p* < 0.01, cluster threshold: *p*-FDR < 0.05. L, left, R, right.

**Figure 5 brainsci-13-01508-f005:**
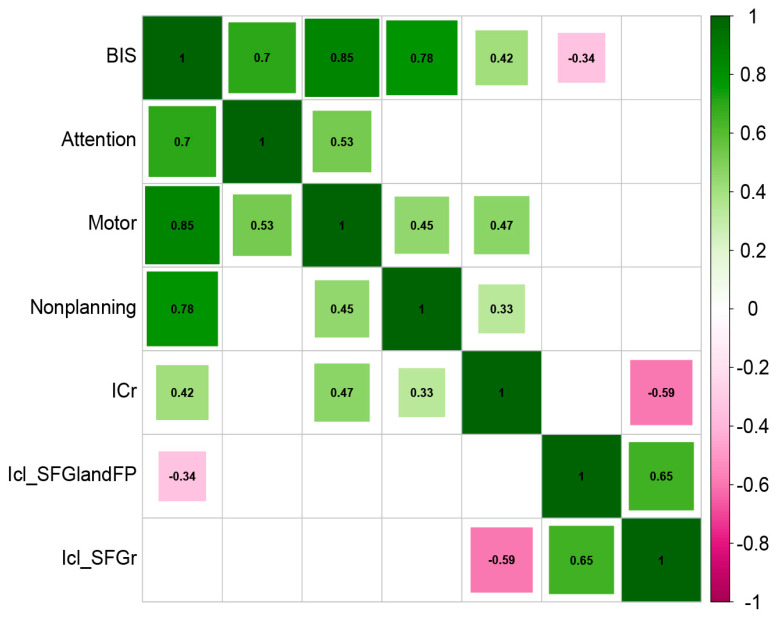
Correlation matrix. Green rectangles represent a positive correlation, while pink represents a negative correlation, and no color means that the correlation had not reached the level of significance. BIS, total BIS-11 score; Attention, score of attention; Motor, score of motor; Nonplanning, score of nonplanning; ICr, FC between the right insula and the lateral occipital cortex and the right angular gyrus; Icl_SFGandFP, FC between the left insula and the left superior frontal gyrus and left frontal pole; Icl_SFGr, FC between the left insula and right superior frontal gyrus.

**Figure 6 brainsci-13-01508-f006:**
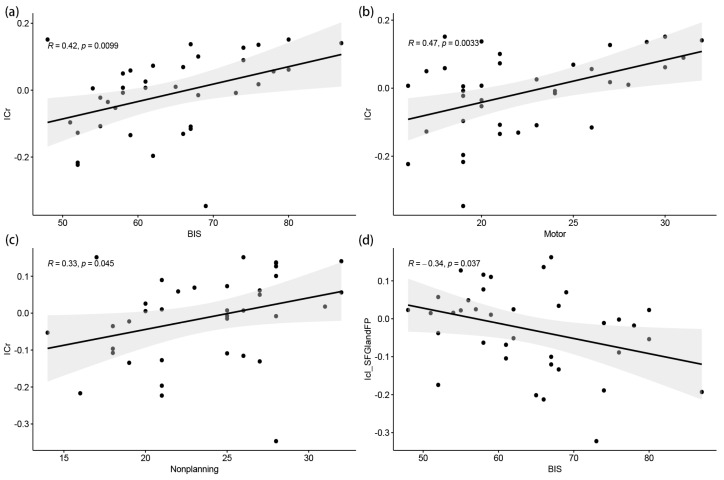
Results of two-by-two related. (**a**) BIS total score was significantly positively correlated with FC between the right insula and the lateral occipital cortex and the right angular gyrus (r = 0.33, *p* = 0.010), horizontal coordinates are behavioral values, vertical coordinates are FC values, lines represent functions fitted to each subject’s data, and gray areas are 95% confidence intervals, same as below; (**b**) motor score was significantly positively correlated with FC between the right insula and the lateral occipital cortex and the right angular gyrus (r = 0.47, *p* = 0.003); (**c**) nonplanning score was significantly positively correlated with FC between the right insula and the lateral occipital cortex and the right angular gyrus (r = 0.33, *p* = 0.045); and (**d**) BIS total score was significantly negatively correlated with FC between the left insula and the left superior frontal gyrus and left frontal pole (r = −0.33, *p* = 0.037). BIS, total BIS-11 score; ICr, FC between the right insula and the lateral occipital cortex and the right angular gyrus; Icl_SFGandFP, FC between the left insula and the left superior frontal gyrus and left frontal pole.

**Table 1 brainsci-13-01508-t001:** Brain regions with differences in FC between the two groups.

ROI	Brain Regions	Volume-Based ROI			Peak *t*-Scores	Voxel Size
		X_(voxel)_	Y_(voxel)_	Z_(voxel)_		
Right insula	sLOC r	20	19	38	4.61	88
	AG r	14	26	36	3.50	45
Left insula	SFG l	72	161	114	−3.72	48
	FP l	39	61	28	−3.57	27
	SFG r	112	157	116	−4.06	17

Note. sLOC r, right superior division lateral occipital cortex; AG r, right angular gyrus; SFG l, left superior frontal gyrus; FP l, left frontal pole; SFG r, right superior frontal gyrus. Voxel threshold: *p* < 0.01, cluster threshold: *p*-FDR < 0.05.

**Table 2 brainsci-13-01508-t002:** Results of regression analysis of insula FC and trait impulsivity.

Predictor Variable	Outcome Variable: Trait Impulsivity				
	*β*	*t*	*p*	*R* ^2^	*F*
FC ICr	0.436	2.162	0.043	0.217	2.770
FC ICl	−0.269	−1.476	0.198		

FC, functional connectivity; ICr, FC between the right insula and the lateral occipital cortex and the right angular gyrus; ICl, FC between the left insula and the left superior frontal gyrus and left frontal pole; *β*, standardized regression coefficient; *t*, *t*-test value; *p*, significance level value; *R*^2^, coefficient of determination; *F*, ANOVA value.

## Data Availability

The data that support the findings of this study are available from the corresponding author, upon reasonable request.

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
