# Peer review of "Insula Connectivity Abnormalities Predict Impulsivity in Chronic Heroin Use Disorder: A Cross-Sectional Resting-State fMRI Study"

_brainsci, 2023, doi:10.3390/brainsci13111508_

Round 1

Reviewer 1 Report

Comments and Suggestions for Authors

Thank you for giving me the opportunity to review this manuscript. I think this manuscript is written well. The following points will make this manuscript more meaningful.

1) It is better to describe clearly that this study is a cross-sectional study in the title, the abstract and the method section.

2) Please describe the hypothesis in the introduction session.

3) Please describe the sample size estimation in the method section.

4) Please describe all predictors, potential confounders and effect modifiers. Please also describe any statistical methods to control for confoundings.

Author Response

Dear Reviewer

Thank you for the comments concerning our manuscript entitled “Abnormalities in insula functional connectivity predict trait impulsivity in patients with chronic heroin use disorder: a resting-state fMRI study” (ID: brainsci-2638771). Those comments are all valuable and very helpful for revising and improving our paper, as well as the important guiding significance to our research. We have studied the comments carefully and have made corrections (modifications are highlighted in red) which we hope meet with approval. In addition, the following are answers to your questions:

R1-1. It is better to describe clearly that this study is a cross-sectional study in the title, the abstract and the method section.

A: Thanks for the comment. As per your suggestion, we have stated that this study is a cross-sectional study in the title (line 3), abstract (line 16), and methods (line 117) sections respectively, and also discussed the shortcomings of this methodology in the limitations section (line 445) as well.

R1-2: Please describe the hypothesis in the introduction session.

A: Thanks for the comment. We added the assumption in the last paragraph of the introduction. We hypothesize that "the heroin group would have higher scores on trait impulsivity and each of its factors than the control group, and that abnormalities in insula FC would be significantly and positively correlated with trait impulsivity scores" (lines 111-114).

R1-3. Please describe the sample size estimation in the method section.

A: Thanks for your careful reading. By looking up the records of this experiment, we added participant selection protocols and sample size estimates (line 117). Our subject selection protocol at the time was the way it was done in the reference literature (Li et al., 2016). However, as of now, our sample size is small, so we also mention it in the limitations section (lines 441-443).

R1-4. Please describe all predictors, potential confounders and effect modifiers. Please also describe any statistical methods to control for confoundings.

A: Thanks for the comment. The main control variables in this study were age (line 221) and parameters generated in the preprocessing (e.g., head movement parameters), and the factors involved in prediction were the functional connectivity values that turned out to be significant after correlation analysis (line 312). In addition, we are using the method of multiple regression analysis to control for confounders (line 221).

Reviewer 2 Report

Comments and Suggestions for Authors

Comments for authors 

The investigators report an examination of the relationship between aspects of impulsivity and insula functional connectivity in a small sample of cases with heroin abuse disorder and in healthy controls. The question is a worthwhile question. The imaging methodology follows well recognised procedures, but the statistical analysis is not adequate, as described below.  They report differing patterns of connectivity of left and right insula. 

However, the findings should be interpreted with caution because: 

  1. The sample size is small 
  2. In light of the sensitivity of FC estimates to movement, removing only the linear effects of head movement is sub-optimal. There is evidence that higher order effects also contribute noise and decrease the quality of the estimate. 
  3. The definition of the insula ROI is not provided. As the Insula is a large region with diverse functional roles, lack of precision in specifying the ROI diminishes the interpretability of the findings 
  4. Lines 237=239 and Figure 5: R insula connectivity is not defined adequately. Do the values refer to connectivity between the right insula and a specific brain regions? If so, this should be made clear.  
  5. The claim that the FC patterns for different aspect of impulsivity is not justified as the investigators did not demonstrate significant differences between the correlation coefficient for different aspect of impulsivity.  Merely demonstrating that a particular pattern of correlation is significantly different for zero for one aspect of impulsivity and but does not reach threshold for significance for the other does not demonstrate that the patterns were significantly different form each other. The investigators should perform the relevant tests. If the differences in correlations are not significant, they should note this weakness in their findings. 

Minor points: 

Valium is not an antipsychotic. 

Line 290:  the relevance of the word somatostatin is unclear.  

Table 1 should include FDR p values 

Comments on the Quality of English Language

Use of English language: 5/5

Author Response

Dear Reviewer

Thank you for the comments concerning our manuscript entitled “Abnormalities in insula functional connectivity predict trait impulsivity in patients with chronic heroin use disorder: a resting-state fMRI study” (ID: brainsci-2638771). Those comments are all valuable and very helpful for revising and improving our paper, as well as the important guiding significance to our research. We have studied the comments carefully and have made corrections (modifications are highlighted in red) which we hope meet with approval. In addition, the following are answers to your questions:

R2-1. The sample size is small.

A: Thanks for the comment. This is a shortcoming of this study. We were unable to obtain a larger sample size due to restricted sampling conditions. We have emphasized this point in the limitations section (lines 441-443).

R2-2. In light of the sensitivity of FC estimates to movement, removing only the linear effects of head movement is sub-optimal. There is evidence that higher-order effects also contribute to noise and decrease the quality of the estimate.

A: Thanks for the comment. We refer to the more standard preprocessing procedure to obtain parameters such as head movement and cerebrospinal fluid and denoise the data (lines 193194), which can mitigate the effect of additional factors to some extent. However, there will still be some bias, so we also present the shortcomings of this method in the limitations section (line 435).

R2-3. The definition of the insula ROI is not provided. As the Insula is a large region with diverse functional roles, a lack of precision in specifying the ROI diminishes the interpretability of the findings.

A: Thanks for your careful reading. We are using the entire insula as the ROI, with the template coming from the atlas in CONN. We added a definition of the ROI (line 218). As you say this ROI is larger, and while it doesn't provide particularly detailed results, it does give some indication of what's going on and is also more in line with the results of previous studies and our hypothesis. We also raise this issue in the limitations section (lines 438-441) and hope to have the opportunity to refine it further in the future.

R2-4. Lines 237=239 and Figure 5: R insula connectivity is not defined adequately. Do the values refer to connectivity between the right insula and specific brain regions? If so, this should be made clear.

A: Thanks for your careful reading. These values refer to the connectivity between the right insula and specific brain regions. At your suggestion, we have detailed these values in the Methods section (lines 208-21o). For example, “FC measures the time-domain correlation between spatially separated brain regions, and the time-series correlation coefficients of brain regions with functional connectivity were significantly higher than those of non-functionally connected brain regions, even though they were spatially separated from each other”.

R2-5. The claim that the FC patterns for different aspects of impulsivity are not justified as the investigators did not demonstrate significant differences between the correlation coefficient for different aspects of impulsivity. Merely demonstrating that a particular pattern of correlation is significantly different for zero for one aspect of impulsivity but does not reach the threshold for significance for the other does not demonstrate that the patterns were significantly different from each other. The investigators should perform the relevant tests. If the differences in correlations are not significant, they should note this weakness in their findings.

A: We appreciate your kind suggestion. By going through the article again, we realized that this was an over-interpretation of our results. As a result, we have revised this part of the description (lines 345-346). We revised it to read "Moreover, changes in right insula FC were also significantly and positively correlated with two-factor structures of trait impulsivity (motor and nonplanning)".

R2-6. Valium is not an antipsychotic.

A: Thanks for your careful reading. It was a mistake in our writing and we deleted the word (line 128).

R2-7. Line 290:  the relevance of the word somatostatin is unclear. 

A: Thanks for your careful reading. It was a mistake in our writing and we deleted the word (line 271).

R2-8. Table 1 should include FDR p-values

A: Thanks for your careful reading. At your suggestion, we have added this to the table notes (line 283).

Reviewer 3 Report

Comments and Suggestions for Authors

This paper by Zhang and colleagues titled ‘Abnormalities in insula functional connectivity predict trait impulsivity in patients with chronic heroin use disorder: a resting-state fMRI study’ investigates the relationship between trait impulsivity and heroin use disorder (HUD) by examining behavioral traits and resting-state functional connectivity (FC) in patients with HUD compared to a control group. The key findings and points of discussion are that patients with HUD exhibit higher levels of total trait impulsivity, as well as specifically in the dimensions of motor impulsivity and nonplanning impulsivity, compared to the control group. Also, enhanced FC is observed between the right insula and regions associated with visual attention, such as the lateral occipital cortex and right angular gyrus, in the HUD group; the left insula shows reduced FC with the left superior frontal gyrus, right superior frontal gyrus, and left frontal pole in the HUD group compared to the control group. Finally, higher sensation-seeking and impulse traits in patients with HUD are associated with enhanced FC in the right insula, indicating a link between these traits and insular function. Changes in FC of the right insula are positively correlated with motor and nonplanning impulsivity, while, in contrast, changes in FC of the left insula, superior frontal gyrus, and frontal pole are negatively correlated with total impulsivity scores, suggesting an influence of impaired inhibitory function on impulsivity.

In summary, this study provides insights into the relationship between trait impulsivity, neural connectivity, and heroin use disorder, highlighting the importance of motor and nonplanning impulsivity in susceptibility to addiction. It also sheds light on lateralized differences in insular function in individuals with addiction and suggests directions for future research. Overall, I find the objective presented in this review article to be quite intriguing, and the authors' insightful observations on this relevant subject matter could capture the attention of Brain Sciences readers. However, there are certain points worth addressing, including specific comments and essential evidence required to bolster the author's argument. These adjustments are necessary to enhance the manuscript's quality, suitability, and overall readability before it can be published in its current state.

-        The title provides a clear overview of the study's focus and methodology. However, there are a few aspects that could be critiqued or refined: it is quite long and contains several complex terms, such as "Abnormalities in Insula Functional Connectivity" and "Trait Impulsivity in Patients with Chronic Heroin Use Disorder." While it is important to be descriptive, a more concise title might be easier for readers to grasp quickly. Also, it is mostly noun-heavy, which can make it less engaging. Adding a verb or action-oriented word could make it more dynamic. For example, "Predicting Trait Impulsivity through Abnormal Insula Connectivity in Chronic Heroin Use Disorder Patients: An fMRI Study" is slightly more action-oriented. A revised title that takes these critiques into account might look like: "Insula Connectivity Abnormalities Predict Impulsivity in Chronic Heroin Use Disorder: An fMRI Study" - it's shorter, more action-oriented, and maintains clarity about the study's focus.

-        I recommend including a graphical abstract that summarizes the main findings of the manuscript.

-        The introduction effectively establishes the significance of the study by explaining the relationship between trait impulsivity and substance use disorders (SUDs). However, it could benefit from a more concise statement of the research's significance and potential contributions to the field. Akso, consider reorganizing the introduction to follow a more logical flow. For example, you could start with the general concept of trait impulsivity, its relevance to addiction, and then progressively narrow down to the specific focus of the study. Finally, to enhance the introduction's comprehensiveness and provide a clearer foundation for the study, I recommend adding a brief section that discusses the neural substrates and brain regions associated with impulsivity and addiction. This addition will help readers understand the neural mechanisms underlying the phenomena being investigated and the importance of studying neural substrates in this context. Specifically, it would be beneficial adding a concise overview of the neural circuits and brain regions associated with impulsivity, with a focus on areas such as the prefrontal cortex, striatum, and insula, as well as an explanation of how understanding neural substrates can provide insights into the mechanisms of addiction, relapse, and the potential targets for intervention and treatment [1-3].

-        Participants: This section provides clear information about the recruitment of participants, including their demographic characteristics. However, it would be beneficial to include more details about the selection process, such as how participants were initially identified as eligible for the study within compulsory drug treatment centers. Additionally, it might be helpful to mention any potential limitations or biases in the selection of participants.

-        The section adequately introduces the research tools used, specifically the BIS-11 questionnaire for measuring impulsivity. However, it might be beneficial to briefly mention the reliability and validity of this instrument, as it is a critical component of the study.

-        The preprocessing steps for the resting-state functional MRI (rs-fMRI) data are outlined in detail, which is valuable for transparency. However, it may be beneficial to provide a concise explanation of why each preprocessing step is necessary and how it helps improve data quality. Additionally, including references to relevant publications or methodologies for each preprocessing step can add credibility to the methodology.

-        The methodology for seed-based analysis is well-described. However, it would be advantageous to briefly explain why the insula was chosen as the region of interest (ROI) for the whole-brain functional connectivity (FC) analysis and how this choice relates to the study's research questions. Additionally, references to relevant literature supporting this choice could strengthen the rationale.

-        Discussion: While the section discusses various findings related to impulsivity and brain functional connectivity in patients with heroin use disorder (HUD), the organization of the content could be improved. It may be helpful to structure the discussion more clearly by separating the key findings into subsections, each addressing a specific aspect of the research.

I hope that, after careful revisions, the manuscript can meet the journal’s high standards for publication. I declare no conflict of interest regarding this manuscript.

Best regards,

Reviewer

References:

1.     https://doi.org/10.1111/acps.13602

2.     https://doi.org/10.3390/biomedicines11051248

1.     DOI: 10.3390/biomedicines11030945

Comments on the Quality of English Language

 Minor editing of English language required.

Author Response

Dear Reviewer

Thank you for the comments concerning our manuscript entitled “Abnormalities in insula functional connectivity predict trait impulsivity in patients with chronic heroin use disorder: a resting-state fMRI study” (ID: brainsci-2638771). Those comments are all valuable and very helpful for revising and improving our paper, as well as the important guiding significance to our research. We have studied the comments carefully and have made corrections (modifications are highlighted in red) which we hope meet with approval. In addition, the following are answers to your questions:

R 3-1. The title provides a clear overview of the study's focus and methodology. However, there are a few aspects that could be critiqued or refined: it is quite long and contains several complex terms, such as "Abnormalities in Insula Functional Connectivity" and "Trait Impulsivity in Patients with Chronic Heroin Use Disorder." While it is important to be descriptive, a more concise title might be easier for readers to grasp quickly. Also, it is mostly noun-heavy, which can make it less engaging. Adding a verb or action-oriented word could make it more dynamic. For example, "Predicting Trait Impulsivity through Abnormal Insula Connectivity in Chronic Heroin Use Disorder Patients: An fMRI Study" is slightly more action-oriented. A revised title that takes these critiques into account might look like this: "Insula Connectivity Abnormalities Predict Impulsivity in Chronic Heroin Use Disorder: An fMRI Study" - it's shorter, more action-oriented, and maintains clarity about the study's focus.

A: Thank you for your guidance on our manuscript. We changed the title at your suggestion to " Insula Connectivity Abnormalities Predict Impulsivity in Chronic Heroin Use Disorder: A Cross-sectional Resting-state fMRI Study" (lines 2-4).

R 3-2. I recommend including a graphical abstract that summarizes the main findings of the manuscript.

A: Thanks for the comment. Following your suggestion, we have included a graphical abstract in the manuscript.

R 3-3. The introduction effectively establishes the significance of the study by explaining the relationship between trait impulsivity and substance use disorders (SUDs). However, it could benefit from a more concise statement of the research's significance and potential contributions to the field. Also, consider reorganizing the introduction to follow a more logical flow. For example, you could start with the general concept of trait impulsivity, its relevance to addiction, and then progressively narrow down to the specific focus of the study. Finally, to enhance the introduction's comprehensiveness and provide a clearer foundation for the study, I recommend adding a brief section that discusses the neural substrates and brain regions associated with impulsivity and addiction. This addition will help readers understand the neural mechanisms underlying the phenomena being investigated and the importance of studying neural substrates in this context. Specifically, it would be beneficial to add a concise overview of the neural circuits and brain regions associated with impulsivity, with a focus on areas such as the prefrontal cortex, striatum, and insula, as well as an explanation of how understanding neural substrates can provide insights into the mechanisms of addiction, relapse, and the potential targets for intervention and treatment [1-3].

A: Thanks for the comment. At your suggestion, we have reorganized the structure of the introduction. To the original, we have added brain structures and functional loops related to trait impulsivity in substance abuse addicts (lines 75-88). In addition, why the insulae were chosen as ROIs was also added (lines 89-103).

R 3-4. Participants: This section provides clear information about the recruitment of participants, including their demographic characteristics. However, it would be beneficial to include more details about the selection process, such as how participants were initially identified as eligible for the study within compulsory drug treatment centers. Additionally, it might be helpful to mention any potential limitations or biases in the selection of participants.

A: Thanks for your careful reading. At your suggestion, we have added detailed information about the selection process by reviewing the experimental records. For example, “All heroin-dependent participants were included in the experimental group after being reviewed by 2 psychiatrists based on their basic information, and physical examination data, and identified through a structured interview” (lines 119-122).

However, we did not document any limitations or biases in the selection of participants at that time, and we will pay closer attention and document this additional information in future studies.

R 3-5. The section adequately introduces the research tools used, specifically the BIS-11 questionnaire for measuring impulsivity. However, it might be beneficial to briefly mention the reliability and validity of this instrument, as it is a critical component of the study.

A: Thanks for the comment. We added the basic information of the scale and calculated its internal consistency reliability coefficient in this study as 0.7, which indicates that this scale has good reliability in this study (line 143).

R 3-6. The preprocessing steps for the resting-state functional MRI (rs-fMRI) data are outlined in detail, which is valuable for transparency. However, it may be beneficial to provide a concise explanation of why each preprocessing step is necessary and how it helps improve data quality. Additionally, including references to relevant publications or methodologies for each preprocessing step can add credibility to the methodology.

A: Thanks for the comment. As you suggested, we have added reasons for using each preprocessing step (such as lines 174-180 and lines 196-197)and added references.

R 3-7. The methodology for seed-based analysis is well-described. However, it would be advantageous to briefly explain why the insula was chosen as the region of interest (ROI) for the whole-brain functional connectivity (FC) analysis and how this choice relates to the study's research questions. Additionally, references to relevant literature supporting this choice could strengthen the rationale.

A: Thanks for the comment. On the question of why is the island of the brain used as an ROI, at your prompting, the relevant explanation has been added in the fifth paragraph of the introduction (lines 89-103), adding to the logic of the introduction.

R 3-8. Discussion: While the section discusses various findings related to impulsivity and brain functional connectivity in patients with heroin use disorder (HUD), the organization of the content could be improved. It may be helpful to structure the discussion more clearly by separating the key findings into subsections, each addressing a specific aspect of the research.

A: Thanks for the comment. At your suggestion, we have divided the main findings into 2 subsections, each of which discusses a specific aspect of the study (characterization and relationships) and also sets a limitations section.

Reviewer 4 Report

Comments and Suggestions for Authors

Firstly, I am writing to express my gratitude for the opportunity to review the research article “Abnormalities in insula functional connectivity predict trait impulsivity in patients with chronic heroin use disorder: a resting-state fMRI study”. I am honored to have been selected to contribute to the peer-review process for Brain Sciences.

I understand the critical importance of rigorous evaluation in academic research and am eager to lend my expertise to this process. I am confident that my analysis will be of value to the authors and help ensure that the work is of the highest quality.

Thank you for entrusting me with this important task. I look forward to the opportunity to provide a thorough and constructive review.

This research delves into impulsivity traits in individuals with heroin use disorder (HUD), highlighting significantly higher levels of overall impulsivity, motor impulsivity, and nonplanning impulsivity compared to a control group. The study utilized neuroimaging to investigate brain connectivity, particularly focusing on the insula region. The findings suggest that the insula may serve as a valuable biomarker for identifying and understanding impulsivity traits in individuals with HUD, shedding light on potential predictive factors for addiction.

I would like to make a series of improvement suggestions to the authors:

INTRODUCCIÓN

  • Clarify the specific impacts of trait impulsivity on addictive behaviors and provide recent statistics or data to emphasize the relevance of understanding impulsivity in the context of heroin use disorder (HUD).

  • Expand on the multidimensional nature of trait impulsivity and its potential relationship with various components of addictive behavior. Provide insights into how different dimensions of impulsivity may contribute to susceptibility to SUDs.

  • Elaborate on recent advancements or debates in research regarding the overall structure and factor components of trait impulsivity, aiming for a more up-to-date perspective and integrating multiple viewpoints on this topic.

  • Enhance the discussion on the neurophysiological basis of trait impulsivity, particularly focusing on recent findings related to abnormalities in brain structure and function, and how these abnormalities may be linked to addictive behaviors in patients with HUD.

  • Include recent evidence highlighting the role of the insula in addiction-related processes and its potential influence on substance use motivation. Emphasize the relevance of investigating insular function and its connection to trait impulsivity to gain a deeper understanding of addictive behaviors.

  • Provide a brief overview of the existing methodologies used in studying impulsivity and functional connectivity (FC) in patients with HUD, with a focus on the strengths and limitations of these approaches to set the stage for the methodology section of the study.

METHOD

2.1. Participants:

  • Provide a rationale for selecting male participants only and discuss potential gender-related implications on impulsivity and addiction.

  • Consider expanding the participant pool to include a more diverse demographic, such as females and individuals with varying cultural backgrounds, to enhance the generalizability of the findings.

  • Include information on socio-economic status or other relevant demographic variables that could influence impulsivity and addiction to better characterize the study participants.

2.3. Magnetic Resonance Imaging (MRI) Data Acquisition:

  • Provide additional details on the specific scanning protocols, such as field strength, sequences used, and any specialized techniques employed in MRI data acquisition.

  • Include information on the duration of each scanning session and any measures taken to ensure participant comfort and minimize motion artifacts during data acquisition.

2.4. RS-fMRI Preprocessing:

  • Offer a more detailed explanation of the rationale behind the preprocessing steps, particularly regarding detrending and regression covariates, to enhance the understanding of the data cleaning process.

  • Discuss potential challenges or limitations associated with preprocessing and provide strategies implemented to mitigate these challenges effectively.

2.5. Seed-based analysis of whole-brain FC:

  • Justify the choice of the insula as the seed region and discuss its relevance to the research question, considering its role in addiction and impulsivity.

  • Discuss potential limitations or alternative approaches in seed-based analysis and justify why this approach was chosen over others.

2.6. Behavioral Data Analysis:

  • Provide a clear explanation of the impulsivity scale scores obtained from the BIS-11 and how these scores were calculated.

  • Describe the statistical tests used to compare impulsivity scores between the HUD and control groups, highlighting any assumptions made and their implications.

  • Consider exploring additional statistical analyses, such as multivariate regression, to investigate the relationship between impulsivity scores and functional connectivity in greater depth.

Overall:

  • Include a discussion on potential limitations and biases associated with the methods used, acknowledging any potential impact on the study's validity and generalizability.

  • Consider suggesting areas for future research or improvements in methodology to address identified limitations and enhance the robustness of the study's findings.

RESULTS

3.1. Behavioral results:

  • Provide a detailed breakdown of the BIS-11 scores for each dimension (attentional impulsivity, motor impulsivity, and nonplanning impulsivity) in both groups to offer a comprehensive view of the impulsivity profiles.

  • Consider incorporating statistical measures of effect size (e.g., partial eta squared) to better convey the magnitude of group differences in impulsivity scores.

3.2. Resting-state FC results:

  • Offer more nuanced insights into the functional significance of the observed differences in insula connectivity by discussing potential implications for cognitive and emotional processing in individuals with HUD.

  • Explore the possible correlations between the FC differences and the duration or severity of heroin use, providing a more comprehensive understanding of how addiction history might influence brain connectivity.

3.3. Association between insula FC and trait impulsivity:

  • Consider discussing the limitations of correlational analyses and regression models, highlighting potential confounding variables and emphasizing the need for caution in making causal inferences.

  • Propose potential neurobiological mechanisms that might underlie the observed relationships between insula FC and trait impulsivity, linking them to existing knowledge about addiction and impulsivity from the neuroscientific literature.

DISCUSSION

Specify Implications for Treatment and Interventions:

  • Discuss how understanding the relationship between impulsivity and brain FC in HUD individuals could inform targeted treatment approaches, such as cognitive-behavioral interventions aimed at modifying impulsivity traits.

  • Explore potential implications for pharmacological interventions targeting specific brain regions related to impulsivity, offering possibilities for more effective addiction treatment.

Explore Longitudinal Studies and Causality:

  • Suggest the need for longitudinal studies to determine the causal relationship between trait impulsivity and substance use in patients with HUD. Discuss the possibility of conducting follow-up assessments to track changes in impulsivity traits over time.

Discuss Limitations and Future Research Directions in Detail:

  • Elaborate on the limitations of the study, such as the small sample size, and propose specific strategies to address these limitations in future research, like multi-site collaborations to increase sample diversity and size.

  • Highlight the need for more comprehensive studies incorporating other potential influencing factors like socio-economic status, childhood experiences, or coexisting mental health conditions, to provide a more holistic understanding of trait impulsivity in HUD patients.

Integrate Neurobiological Mechanisms:

  • Delve deeper into the neurobiological underpinnings of impulsivity and addiction, discussing how alterations in insula connectivity may relate to dopamine pathways or other neurotransmitter systems implicated in addiction and impulsive behavior.

Consider Practical Applications:

  • Discuss how the findings could be translated into real-world applications, such as in the development of targeted addiction prevention programs or public health campaigns aimed at reducing impulsivity-related risk factors for drug addiction.

CONCLUSION

Integration of Factor Structure:

  • Discuss the importance of further investigating and understanding the factor structure of trait impulsivity in patients with HUD, emphasizing its role in addiction vulnerability and treatment strategies.

Longitudinal and Interventional Studies:

  • Propose the need for longitudinal studies to track changes in impulsivity traits over time and assess how they correlate with addiction severity and relapse rates. Additionally, suggest interventional studies to determine the potential of modifying trait impulsivity to improve addiction outcomes.

Multi-Modal Approaches:

  • Suggest incorporating a multimodal approach by integrating various neuroimaging techniques (e.g., fMRI, DTI) along with behavioral assessments to gain a comprehensive understanding of impulsivity-related neurobiological changes in HUD patients.

Gender and Age Considerations:

  • Encourage future research to consider gender and age differences in trait impulsivity within the context of heroin addiction, which might provide nuanced insights into addiction mechanisms and treatment approaches tailored to specific demographics.

Functional Connectivity Alterations:

  • Further investigate the functional connectivity alterations beyond the insula, exploring the interconnected brain regions to unveil a more comprehensive network involved in impulsivity in patients with HUD.

Linking to Treatment Strategies:

  • Discuss the potential implications of the identified insular biomarkers for developing targeted therapeutic interventions, such as neuromodulation techniques or cognitive training, to address impulsivity and enhance addiction treatment outcomes.

Typographical Errors:

  • Please review the naming of Figure 5, as there are 2 figures labeled "Figure 5".

  • Throughout the document, the term "insulae" is used, and I believe it is incorrect; it should be "insula".

Continue the excellent effort; your commitment and diligence are clearly reflected in the caliber of your research. I am confident that, with some refinements, this manuscript will be prepared for submission.

Reviewing your work has been a delightful experience, and I am assured that, with the proposed revisions, your paper will significantly enrich the field. I extend my best wishes for your ongoing research endeavors and eagerly anticipate your forthcoming publications.

I'd like to convey my heartfelt gratitude for the dedication and hard work you've put into your research. The article stands to benefit from substantial enhancements, and I strongly recommend that the authors undertake a comprehensive revision to elevate its quality before submitting it again.

Regards,

Author Response

Dear Reviewer

Thank you for the comments concerning our manuscript entitled “Abnormalities in insula functional connectivity predict trait impulsivity in patients with chronic heroin use disorder: a resting-state fMRI study” (ID: brainsci-2638771). Those comments are all valuable and very helpful for revising and improving our paper, as well as the important guiding significance to our research. We have studied the comments carefully and have made corrections (modifications are highlighted in red) which we hope meet with approval. In addition, the following are answers to your questions:

R 4-1. Clarify the specific impacts of trait impulsivity on addictive behaviors and provide recent statistics or data to emphasize the relevance of understanding impulsivity in the context of heroin use disorder (HUD).

A: Thanks for the comment. At your suggestion, we combed through the results of current research with animal and human participants (lines 44-50), emphasizing the importance of understanding the correlation between impulsivity and HUD.

R 4-2. Expand on the multidimensional nature of trait impulsivity and its potential relationship with various components of addictive behavior. Provide insights into how different dimensions of impulsivity may contribute to susceptibility to SUDs.

A: Thanks for the comment. We emphasize the importance of expanding the multidimensionality of trait impulsivity by citing both previous calls for attention to the factor structure of trait impulsivity (such as Stanford et al., 2009) and the clinical implications of the argument (line 54). However, due to the large number of studies on the dimensionality of trait impulsivity, we primarily use the most extensive three-factor structure to develop our analysis and discussion (lines 58-59).

Stanford, M.S.; Mathias, C.W.; Dougherty, D.M.; Lake, S.L.; Anderson, N.E.; Patton, J.H. Fifty Years of the Barratt Impulsiveness Scale: An Update and Review. Personal. Individ. Differ. 2009, 47, 385–395, doi:10.1016/j.paid.2009.04.008.

R 4-3. Elaborate on recent advancements or debates in research regarding the overall structure and factor components of trait impulsivity, aiming for a more up-to-date perspective and integrating multiple viewpoints on this topic.

A: Thanks for the comment. We combed through current research on the relationship between different factors of trait impulsivity and addictive behaviors (lines 65-72), but there is no current uniformity of results (lines 72-74), which therefore became one of the reasons why we conducted the present study.

R 4-4. Enhance the discussion on the neurophysiological basis of trait impulsivity, particularly focusing on recent findings related to abnormalities in brain structure and function, and how these abnormalities may be linked to addictive behaviors in patients with HUD.

A: Thanks for the comment. At the suggestion of you and other reviewers, we have added a discussion of the neurophysiologic basis of trait impulsivity in the fourth paragraph of the Introduction. This makes our introduction more logical.

R 4-5. Include recent evidence highlighting the role of the insula in addiction-related processes and its potential influence on substance use motivation. Emphasize the relevance of investigating insular function and its connection to trait impulsivity to gain a deeper understanding of addictive behaviors.

A: Thanks for the comment. At your suggestion, we add the latest evidence on the role of insulae in addiction-related processes and their potential impact on motivation for substance use (lines 89-103). We find that " insula can modulate substance use motivation through direct projections to the ventral striatum and orbitofrontal cortex (OFC)" and we therefore use the insula as an ROI.

R 4-6. Provide a brief overview of the existing methodologies used in studying impulsivity and functional connectivity (FC) in patients with HUD, with a focus on the strengths and limitations of these approaches to set the stage for the methodology section of the study.

A: Thanks for the comment. At your suggestion, we added in the last paragraph of the introduction the reason why we chose fMRI because of its high spatial resolution (lines 104-105). In addition, we also present, in the Methods section, the reasons for choosing the seed-based analysis approach (lines 210-214).

R 4-7. Provide a rationale for selecting male participants only and discuss potential gender-related implications on impulsivity and addiction.

A: Thanks for the comment. Due to sampling conditions, we were only able to collect information from male HUD participants, so all participants in this study were set to male. However, this group is not representative of the entire HUD population, so we also emphasize this shortcoming in the limitations section.

R4-8. Consider expanding the participant pool to include a more diverse demographic, such as females and individuals with varying cultural backgrounds, to enhance the generalizability of the findings.

A: Thanks for the comment. We emphasized in the limitations section the need to expand the range of participants in future studies. We wrote that “First, it has been shown that gender factors play an important role in impulsive and risky behaviors [71], but in this study, due to the sampling condition, our participants were all male, which means the generalizability of the results will be affected. Future studies should consider expanding the participant pool to include a more diverse demographic, such as females and individuals with varying cultural backgrounds, to enhance the generalizability of the findings.”

R4-9. Include information on socio-economic status or other relevant demographic variables that could influence impulsivity and addiction to better characterize the study participants.

A: Thanks for the comment. We added one demographic piece of information: daily intake, but were unable to provide this information because we had not previously focused on socioeconomic status information. We discuss this shortcoming in the limitations section.

R4-10. Provide additional details on the specific scanning protocols, such as field strength, sequences used, and any specialized techniques employed in MRI data acquisition.

A: Thanks for the comment. We have added some detailed information about the scanning program. We wrote that “Conventional anatomical images were performed using a small-angle excitation fast gradient echo sequence (T1WI-FLAIR) with transverse axial scans covering the whole brain.” (lines 153-156)

R4-11. Include information on the duration of each scanning session and any measures taken to ensure participant comfort and minimize motion artifacts during data acquisition.

A: Thanks for the comment. We have added a section on experimental procedures to illustrate specific “experimental procedures” with information such as timing (lines 144-150). We wrote that “2.3. Experimental Procedure Before the experiment, participants completed the Barratt Impulsiveness Scale and the Personal Information Questionnaire. Then, after ensuring that the participants are in good health and meet the safety standards of the MRI experiment, they are escorted by a trained scanner into the MRI suite to complete the resting state scan with their eyes closed. The body, especially the head, is kept still throughout the scan. The duration of the scan is approximately 10 minutes.”

2.4. RS-fMRI Preprocessing:

R4-12. Offer a more detailed explanation of the rationale behind the preprocessing steps, particularly regarding detrending and regression covariates, to enhance the understanding of the data-cleaning process.

A: Thanks for the comment. At your suggestion, we have added the rationale behind some of the preprocessing steps by reviewing the references, which are also attached as citations. For example, “In order to control the effect of head movements on the signal due to fatigue or respiration and blood pulsation, which are inevitable for participants during the scanning process, the head movement correction algorithm performs model alignment for rotation around the coordinate axis and translation in the direction of the coordinate axis, so that the anatomical position of the brain corresponding to a given pixel point is the same at different moments of time, even if the images at each point in time are aligned [40].”

R4-13. Discuss potential challenges or limitations associated with preprocessing and provide strategies implemented to mitigate these challenges effectively.

A: Thanks for the comment. As you say, current pretreatment is not completely accurate, and this study, while choosing a more mature pretreatment scheme, still has shortcomings, so we have also added a discussion of this issue in the limitations section.

2.5. Seed-based analysis of whole-brain FC:

R4-14. Justify the choice of the insula as the seed region and discuss its relevance to the research question, considering its role in addiction and impulsivity.

A: Thanks for the comment. We have followed your and other reviewers' suggestions and have presented in the introduction section the reasons why brain islands were chosen as ROIs (lines 95-98). We wrote that “The insula is a key brain region that supports endoreception and is also involved in ad-diction-related processes [30]. It has been found that the insula can modulate substance use motivation through direct projections to the ventral striatum and orbitofrontal cortex (OFC) [31,32].”

R4-15. Discuss potential limitations or alternative approaches in seed-based analysis and justify why this approach was chosen over others.

A: Thanks for the comment. In conjunction with the purpose of our study, the current research methodology is somewhat more appropriate. We also added about why we used the current research approach (lines 207-214). We wrote that “FC measures the time-domain correlation between spatially separated brain regions, and the time-series correlation coefficients of brain regions with functional connectivity were significantly higher than those of non-functionally connected brain regions, even though they were spatially separated from each other [45]. The seed-based analysis is essentially a model-based approach where we can select a seed point or ROI and find the linear correlation of that seed-point region with all other voxels throughout the brain to obtain a seed-based FC map. The simplicity, interpretability, and directness of this technique make it an excellent method to use for studying rs-fMRI FC [45].”

The method is introduced

2.6. Behavioral Data Analysis:

R4-16. Provide a clear explanation of the impulsivity scale scores obtained from the BIS-11 and how these scores were calculated.

A: Thanks for the comment. Based on your suggestions, we have added detailed information to the scale introduction session, such as how it is scored and what the reliability and validity are (lines 140-143).

R4-17. Describe the statistical tests used to compare impulsivity scores between the HUD and control groups, highlighting any assumptions made and their implications.

A: Thanks for the comment. The independent samples t-test used in this study was used for hypothesis testing only since the data satisfied normal distribution (lines 233-234). In addition, we added Bayesian factors (lines 237-246), thus making the results of the study more reliable.

R4-18. Consider exploring additional statistical analyses, such as multivariate regression, to investigate the relationship between impulsivity scores and functional connectivity in greater depth.

A: Thanks for the comment. At your suggestion, we added multiple regression analyses to examine the relationship between impulsivity scores and FC in more depth and to highlight the methodology of this analysis (lines 249 and 312).

Overall:

Include a discussion on potential limitations and biases associated with the methods used, acknowledging any potential impact on the study's validity and generalizability.

Consider suggesting areas for future research or improvements in methodology to address identified limitations and enhance the robustness of the study's findings.

3.1. Behavioral results:

R4-19. Provide a detailed breakdown of the BIS-11 scores for each dimension (attentional impulsivity, motor impulsivity, and nonplanning impulsivity) in both groups to offer a comprehensive view of the impulsivity profiles.

A: Thanks for the comment. We refer to the more established three-factor model, and no further breakdown based on this three-factor model was found, so a detailed breakdown of BIS-11 scores for each of the dimensions (attention, movement, and unplanned impulsivity) in the two groups is not available at this time.

R4-20. Consider incorporating statistical measures of effect size (e.g., partial eta squared) to better convey the magnitude of group differences in impulsivity scores.

A: Thanks for the comment. The independent samples t-test was used in this study, and Cohen's d may be more suitable for this statistical method (Correll et al., 2020), therefore, Cohen's d was used as the effect size in this study.

Correll, J., Mellinger, C., McClelland, G. H., & Judd, C. M. (2020). Avoid Cohen's 'Small', 'Medium', and 'Large' for Power Analysis. Trends in cognitive sciences, 24(3), 200–207. https://doi.org/10.1016/j.tics.2019.12.009

3.2. Resting-state FC results:

R4-21. Offer more nuanced insights into the functional significance of the observed differences in insula connectivity by discussing potential implications for cognitive and emotional processing in individuals with HUD.

A: Thanks for the comment. We apologize that we did not consider the role of emotional factors, which is a shortcoming of this study, and therefore we discuss this issue in the limitations section.

Not processed, incorporated into limitations

R4-22. Explore the possible correlations between the FC differences and the duration or severity of heroin use, providing a more comprehensive understanding of how addiction history might influence brain connectivity.

A: Thanks for the comment. Because the linear correlation between the duration of drug use and impulsivity was not significant, we did not include it in our multiple regression modeling.

Not processed, incorporated into limitations

3.3. Association between insula FC and trait impulsivity:

R4-23. Consider discussing the limitations of correlational analyses and regression models, highlighting potential confounding variables and emphasizing the need for caution in making causal inferences.

A: Thanks for the comment. Because the correlation analysis and regression models used in this study do not account for causality, the discussion in this study also emphasizes "predictive" rather than causal relationships. We have also emphasized the shortcomings of this method in the limitations section.

R4-25. Propose potential neurobiological mechanisms that might underlie the observed relationships between insula FC and trait impulsivity, linking them to existing knowledge about addiction and impulsivity from the neuroscientific literature.

A: Thanks for the comment. We propose potential neurobiological mechanisms in the relational link between brain islands and impulsivity. For example, “The angular gyrus is located in the posterior part of the inferior parietal lobe, where multisensory (auditory, visual, and somatosensory) information converges and is asso-ciated with multiple cognitive domains such as memory [56] and visuospatial attention [58,59]. From a functional brain perspective, this result demonstrates that patients with HUD may have higher sensation-seeking [12,59]”. This potential mechanism can explain our results and can also provide some insights for future mechanism studies.

Specify Implications for Treatment and Interventions:

R4-26. Discuss how understanding the relationship between impulsivity and brain FC in HUD individuals could inform targeted treatment approaches, such as cognitive-behavioral interventions aimed at modifying impulsivity traits.

A: Thanks for the comment. Because we understand the relationship between impulsivity and brain function in HUD patients, it may be possible in the future to treat addictive behaviors through neuromodulation techniques that target brain islands. Based on your suggestion, we have added this to the discussion section (lines 424-426).

R4-27. Explore potential implications for pharmacological interventions targeting specific brain regions related to impulsivity, offering possibilities for more effective addiction treatment.

A: Thanks for the comment. We believe that the present study suggests the possibility of treating addictive behaviors in the future through neuromodulation techniques that target brain islands (lines 424-426).

R4-28. Explore Longitudinal Studies and Causality: Suggests the need for longitudinal studies to determine the causal relationship between trait impulsivity and substance use in patients with HUD. Discuss the possibility of conducting follow-up assessments to track changes in impulsivity traits over time.

A: Thanks for the comment. We added in the limitations section that future causality results can be obtained through longitudinal studies.

Discuss Limitations and Future Research Directions in Detail:

R4-29. Elaborate on the limitations of the study, such as the small sample size, and propose specific strategies to address these limitations in future research, like multi-site collaborations to increase sample diversity and size.

A: Thanks for the comment. We discussed the small sample size as a limitation, and we also referenced your suggestions to add meticulous ways for future research to compensate. We wrote that “the sample size of this study was small and future studies could be conducted through multi-site collaborations to increase the diversity and size of the sample.”

R4-30. Highlight the need for more comprehensive studies incorporating other potential influencing factors like socio-economic status, childhood experiences, or coexisting mental health conditions, to provide a more holistic understanding of trait impulsivity in HUD patients.

A: Thanks for the comment. We have added a discussion of the lack of control of additional factors at the suggestion of you and other reviewers. We wrote that “factors such as socio-economic status, emotion, and parameter settings of pre-processing also affect research reliability and validity, and further optimization of the experimental process is also needed in the future to exclude the interference of irrelevant factors as much as possible.”

R4-31. Integrate Neurobiological Mechanisms: Delve deeper into the neurobiological underpinnings of impulsivity and addiction, discussing how alterations in insula connectivity may relate to dopamine pathways or other neurotransmitter systems implicated in addiction and impulsive behavior.

A: Thanks for the comment. We have added a description of the underlying neural mechanisms in the Discussion section (lines 422-424), and we have also added future research on this direction in the Limitations section at your suggestion. We wrote that “Future intervention or longitudinal studies are needed to investigate the causal relationship and neurobiological mechanisms (such as dopamine pathway) between these two factors.”

R4-32. Consider Practical Applications: Discuss how the findings could be translated into real-world applications, such as in the development of targeted addiction prevention programs or public health campaigns aimed at reducing impulsivity-related risk factors for drug addiction.

A: Thanks for the comment. In response to your suggestion, we have added a discussion section on the application of results (lines 426-427). We wrote that “It also helps us to develop targeted addiction prevention programs or public health campaigns to reduce risk factors for addiction related to impulsivity”

CONCLUSION

R4-33. Integration of Factor Structure: Discuss the importance of further investigating and understanding the factor structure of trait impulsivity in patients with HUD, emphasizing its role in addiction vulnerability and treatment strategies.

A: Thanks for the comment. We have added what you suggested in the conclusion section. We wrote that “By focusing on the structure of trait impulsivity factors, it helps to accurately predict and personnel subgroup levels.”

R4-34. Longitudinal and Interventional Studies: Propose the need for longitudinal studies to track changes in impulsivity traits over time and assess how they correlate with addiction severity and relapse rates. Additionally, suggest interventional studies to determine the potential of modifying trait impulsivity to improve addiction outcomes.

A: Thanks for the comment. In conjunction with your previous suggestions, we have included a discussion of longitudinal and intervention studies that could be undertaken in the future in the Limitations and Research Perspectives section (lines 447-451). We wrote that “Future intervention or longitudinal studies are needed to investigate the causal relationship and neurobiological mechanisms (such as dopamine pathways) between these two factors. For example, a follow-up assessment was conducted to track changes in HUD trait impulsivity over time and its dynamic relationship with insula and addictive behaviors.”

R4-35. Multi-Modal Approaches: Suggest incorporating a multimodal approach by integrating various neuroimaging techniques (e.g., fMRI, DTI) along with behavioral assessments to gain a comprehensive understanding of impulsivity-related neurobiological changes in HUD patients.

A: Thanks for the comment. In conjunction with your previous suggestions, we have included a discussion of longitudinal and intervention studies that could be undertaken in the future in the Limitations and Research Perspectives section (lines 447-451).

R4-36. Gender and Age Considerations: Encourage future research to consider gender and age differences in trait impulsivity within the context of heroin addiction, which might provide nuanced insights into addiction mechanisms and treatment approaches tailored to specific demographics.

A: Thanks for the comment. We included age as a control variable. However, we only selected male participants, so we discuss this shortcoming in the limitations section. We wrote that “it has been shown that gender factors play an important role in impulsive and risky behaviors [71], but in this study, due to the sampling condition, our participants were all male, which means the generalizability of the results will be affected. Future studies should consider expanding the participant pool to include a more diverse demographic, such as females and individuals with varying cultural backgrounds, to enhance the generalizability of the findings.”

R4-37. Functional Connectivity Alterations: Further investigate the functional connectivity alterations beyond the insula, exploring the interconnected brain regions to unveil a more comprehensive network involved in impulsivity in patients with HUD.

A: Thanks for the comment. This study focuses on the insula, a brain region. Based on your suggestions, we propose in the Limitations and Future Perspectives section that we could focus on other brain regions in the future (lines 449-451). We wrote that “For example, a follow-up assessment was conducted to track changes in HUD trait im-pulsivity over time and its dynamic relationship with insula (or other brain regions FC) and addictive behaviors.”

R4-38. Linking to Treatment Strategies: Discuss the potential implications of the identified insular biomarkers for developing targeted therapeutic interventions, such as neuromodulation techniques or cognitive training, to address impulsivity and enhance addiction treatment outcomes.

A: Thanks for the comment. We believe that the insula can be used as a target for neuromodulation techniques to treat or intervene in addictive behaviors (lines 424-426).

Typographical Errors:

R4-39. Please review the naming of Figure 5, as there are 2 figures labeled "Figure 5".

A: Thanks for careful reading. We have fixed this markup error.

R4-40. Throughout the document, the term "insulae" is used, and I believe it is incorrect; it should be "insula".

A: Thanks for careful reading. We have fixed this error word.

Round 2

Reviewer 2 Report

Comments and Suggestions for Authors

The authors have addressed several of my concerns adequately.

However I remain concerned about the meaning of the Functional connectivity of  ICr in figure 5, row 5; in figure.6 a,b,and c; and in lines 299-301 of the text

Normally functional connectivity is measured between pairs of brain regions. When the authors refer to the functional connectivity of  ICr to they mean the average functional connectivity of ICr with all of the brain regions with which it exhibits significant connectivity or do they mean functional connectivity with one specific brain region.

Author Response

Dear Reviewer

The new round of comments you have provided have been very helpful in revising and improving our paper, and have been an important guide to our research. We have carefully studied your comments and made changes (modifications are highlighted in yellow), which we hope will be approved by you. In addition, here are the answers to the questions you have asked:

R2-1. However, I remain concerned about the meaning of the Functional connectivity of ICr in figure 5, row 5; in figure.6 a, b, and c; and in lines 299-301 of the text

Normally functional connectivity is measured between pairs of brain regions. When the authors refer to the functional connectivity of ICr to they mean the average functional connectivity of ICr with all of the brain regions with which it exhibits significant connectivity or do they mean functional connectivity with one specific brain region.

A: We apologize for not making accurate changes initially due to our failure to understand your comments. At your suggestion, we have carefully revised the description of FC. We had previously simply written the functional connection between the right insula and other brain regions as "FC right insula", which also led to a lot of ambiguity. Therefore, we have rewritten it as "FC between the right insula and the lateral occipital cortex and the right angular gyrus". We checked the entire manuscript (including the abstract) and corrected the description. For example, in lines 20-21, 281-283, 310, 321-323, 326-328 and 330-333.

Reviewer 3 Report

Comments and Suggestions for Authors

Dear Authors,

I am pleased to acknowledge that you have indeed addressed all of my concerns and queries in a clear and precise manner. Your responses have provided valuable insights into the modifications made to the manuscript in light of my comments. It is evident that you have taken great care to ensure that the revised manuscript aligns more closely with the scientific rigor expected for publication in Brain Sciences.

Upon reviewing the updated version, I find that the inclusion of the additional studies has indeed enriched the understanding of neural substrates and brain regions associated with impulsivity and addiction. The provided studies contribute significantly to the comprehensiveness of the section. However, in order to provide a more holistic view of the neural structures, I believe there's still an opportunity to expand upon certain factors. Specifically, the discussion of the neural mechanisms underlying the phenomena being investigated could offer a deeper insight into the mechanisms at play (https://doi.org/10.1111/acps.13602; https://doi.org/10.3390/biomedicines11051248; DOI: 10.3390/biomedicines11030945). 

I want to reiterate my appreciation for your responsiveness and willingness to consider these suggestions. I believe that this minor revision will significantly enhance the quality and impact of the Introduction section. 

Thank you once again for your dedication to improving the manuscript. I look forward to seeing the continued progress.

Best regards,

Reviewer

Author Response

Dear Reviewer

Thank you for the comments concerning our manuscript entitled “Insula Connectivity Abnormalities Predict Impulsivity in Chronic Heroin Use Disorder: A Cross-sectional Resting-state fMRI Study” (ID: brainsci-2638771). The new round of comments you have provided have been very helpful in revising and improving our paper, and have been an important guide to our research. We have carefully studied your comments and made changes (modifications are highlighted in yellow), which we hope will be approved by you. In addition, here are the answers to the questions you have asked:

R4-1. However, in order to provide a more holistic view of the neural structures, I believe there's still an opportunity to expand upon certain factors. Specifically, the discussion of the neural mechanisms underlying the phenomena being investigated could offer a deeper insight into the mechanisms at play (https://doi.org/10.1111/acps.13602; https://doi.org/10.3390/biomedicines11051248; DOI: https://doi.org/10.3390/biomedicines11030945).

A: Thanks for the comment. We reviewed the papers you provided, and in conjunction with the content of these articles, we reviewed neurobiology papers related to this article to enrich our understanding of the neural substrates and brain regions associated with impulsivity and addiction in the Introduction and Discussion sections.

In the Introduction section (lines 96-99), we added how brain insulae are involved in the motivation-habit pathway, thus providing a stronger theoretical basis for why we chose them as ROIs. We wrote that “Results from Deng and colleagues pinpoint specific functions of the anterior insular cortex → the brainstem nucleus tractus solitarii circuit for selectively controlling motivational vigor and suggest that motivation is in part regulated by top-down regulation of dopamine signaling by the insula anterior cortex [33].”

In the discussion section (lines 435-439), we explain our findings and their underlying mechanisms by eliciting 5 hydroxytryptophan. We wrote that “Numerous studies have found that 5-hydroxytryptamine (5-HT) in the prefrontal cortex plays an important role in regulating an individual's impulsivity [66,67] and that elevated concentrations of 5-HT reduce the level of compulsive cocaine-seeking in mice, which in turn prevents the onset of addiction [68]. This further supports our findings, suggesting that trait impulsivity may precede addiction [18]”

In addition (lines 467-469), we propose in the limitations section that " We also need to further explore, at the molecular level, the biomarkers and neurobiological basis of trait impulsivity, thus providing a more comprehensive neurostructural perspective [68]"